# Machine-learning models of $\delta^{13} C$ and $\delta^{15} N$ isoscapes in Amazonian wood

Isabela M. Souza-Silva<sup>1,2</sup>, Luiz A. Martinelli<sup>1</sup>, Brent Holmes<sup>2</sup>, Ana C. G. Batista<sup>1</sup>, Maria G. S. Araújo<sup>1</sup>, Anna L. Garção<sup>1</sup>, Stéphane Ponton<sup>3</sup>, Peter Groenendijk<sup>4</sup>, Giuliano M. Locosselli<sup>1</sup>, Daigard R. Ortega-Rodriguez<sup>5</sup>, Deoclecio J. Amorim<sup>1</sup>, Fábio J. V. Costa<sup>6</sup>, Gabriela B. Nardoto<sup>7</sup>, Alexandre T. Brunello<sup>1</sup>, Vladimir Eliodoro Costa<sup>8</sup>, Gabriel Assis-Pereira<sup>9</sup>, Mario Tomazello-Filho<sup>5</sup>, Niro Higuchi<sup>10</sup>, Ana C. Barbosa<sup>11</sup>, João Paulo Sena-Souza<sup>12</sup>, and Clément P. Bataille<sup>2,13</sup>

Correspondence: Isabela M. Souza-Silva (isabela.aliam@gmail.com) and Clément P. Bataille (cbataill@purdue.edu)

Abstract. Illegal logging is one of the most prevalent environmental infractions in the Amazon, led by organized networks that cause substantial ecological and economic impacts. Official control mechanisms, such as Brazil's Forest Origin Document (DOF), remain vulnerable to the fraudulent manipulation of virtual timber credits and inconsistencies in digital traceability. These deficiencies highlight the need for independent, scientifically based methodologies for timber traceability that can support law enforcement and ensure reliable provenance verification. Here, we tested whether the isotopic composition of carbon ( $\delta^{13}$ C) and nitrogen ( $\delta^{15}$ N) in wood can trace Amazonian timber origin. We developed basin-wide  $\delta^{13}$ C and  $\delta^{15}$ N isoscapes using machine-learning models to predict spatial variability. A total of 571 trees from 47 sites were analyzed for both isotopes. Tree disks or wedges were sampled from the basal trunk, sectioned transversely, and sub-sampled from heartwood to near the sapwood boundary to obtain whole-tree isotopic composition. The  $\delta^{13}$ C and, more strongly, the  $\delta^{15}$ N values exhibited substantial within-site heterogeneity, indicating individual-level physiological controls, interspecific differences, and/or small-scale environmental variation influencing isotope fractionation. Despite these sources of noise, isotopic values showed independent and predictable spatial patterns across the basin ( $R^2 = 0.67$  for  $\delta^{15}$ N and  $R^2 = 0.60$  for  $\delta^{13}$ C). Nitrogen isotopes

<sup>&</sup>lt;sup>1</sup>Center for Nuclear Energy in Agriculture, University of São Paulo, Av. Centenário 303, Piracicaba, SP 13416-000, Brazil

<sup>&</sup>lt;sup>2</sup>Department of Earth and Environmental Sciences, University of Ottawa, 25 Templeton St., Ottawa, ON K1N 6N5, Canada

<sup>&</sup>lt;sup>3</sup>SILVA UMR 1434, INRAE, Université de Lorraine, AgroParisTech, F-54280 Champenoux, France

<sup>&</sup>lt;sup>4</sup>Department of Plant Biology, Institute of Biology, University of Campinas, Campinas, SP, Brazil

<sup>&</sup>lt;sup>5</sup>Department of Forest Sciences, Luiz de Queiroz College of Agriculture, University of São Paulo, Av. Pádua Dias 11, Piracicaba, SP 13418-900, Brazil

<sup>&</sup>lt;sup>6</sup>National Institute of Criminalistics, Federal Police, Setor Policial Sul, Lote 7, Asa Sul, Brasília, DF 70610-902, Brazil

<sup>&</sup>lt;sup>7</sup>Department of Ecology, Institute of Biological Sciences, University of Brasília, Campus Universitário Darcy Ribeiro Bloco E, Brasília, DF 70910-900, Brazil

<sup>&</sup>lt;sup>8</sup>Stable Isotopes Center, São Paulo State University (UNESP), Prof. Dr. Antônio Celso Wagner Zanin Street 250, Rubião Junior, Botucatu, SP 18618-689, Brazil

<sup>&</sup>lt;sup>9</sup>Research and Development Department, Monte Verde Carbon, Itajubá, Minas Gerais, Brazil

<sup>&</sup>lt;sup>10</sup>National Institute for Amazon Research, Av. André Araújo 2936, Petrópolis, Manaus, AM 69067-375, Brazil

<sup>&</sup>lt;sup>11</sup>Department of Forest Sciences, University of Lavras, P.O. Box 3037, Lavras, MG 37200-000, Brazil

<sup>&</sup>lt;sup>12</sup>Department of Geosciences, State University of Montes Claros (Unimontes), Professor Darcy Ribeiro Campus, Montes Claros, MG 39401-089, Brazil

<sup>&</sup>lt;sup>13</sup>Department of Forestry and Natural Resources, Purdue University, West Lafayette, IN 47907, USA

were primarily controlled by edaphic factors, while carbon isotopes revealed a broad longitudinal gradient linked to climate. Together, these isotopic markers provide complementary information for basin-scale timber provenancing and form a robust, high-resolution framework for Amazon-wide traceability.

## 1 Introduction

The Amazon rainforest plays a critical role in regulating climate processes on both continental and global levels, with moisture recycling and vapor fluxes that influence rainfall patterns from the Atlantic Ocean to the Andes and to the agricultural regions of south-central Brazil (Salati et al., 1979). Notwithstanding its significance, the region has experienced a loss of more than one-fifth of its forest cover since the 1980s, when the coverage index was approximately 95% (INPE, 2023). The conversion of forested areas continues to accelerate, frequently linked to land grabbing and the expansion of agriculture (Nepstad et al., 2014; Matricardi et al., 2020; Ferrante et al., 2021; Silva Junior et al., 2020), with illegal logging frequently preceding and facilitating subsequent changes in land use (Saraiva, 2021).

Globally, it is estimated that between 15% and 30% of timber traded is illegal, with this proportion potentially reaching 50% to 90% in tropical regions (INTERPOL, 2019). In the Amazon, the scale of illegal logging has become increasingly evident. Data from Rede SIMEX indicate that, from August 2020 to July 2021, roughly 40% of timber extraction took place without authorization, largely on private properties (72%) and, to a lesser extent, on Indigenous lands and protected areas (15%) (Valdiones et al., 2022). In this context, governmental strategies to control illegality, such as in Brazil, have advanced with the approval of the Non-Prejudicial Extraordinary Report (NDF), which certifies the legality and sustainability of timber exploitation for species such as *cumaru* (*Dipteryx* spp.), *ipê* (*Handroanthus*, *Tabebuia*, *Roseodendron*) and pink cedar (*Cedrela*), included in Appendix II of CITES (IBAMA, 2024). However, deficiencies in governance still exacerbate the vulnerability of other species through management plans that overestimate harvestable volumes and systematically misidentify valuable taxa, thereby facilitating the "legalization" of timber of illicit origin (Brancalion et al., 2018; CNI, 2018).

In Brazil, the system for formal provenance control, grounded in the Forest Origin Document (DOF – *Documento de Origem Florestal*) and Forest Guides (*Guias Florestais*), has limited legal value; it is primarily documentary in nature and does not enable enforcement of regulations or best practices to combat illegal logging. Even when those DOF practices are followed, inconsistencies may emerge between registration and load throughout the logistics chain, thereby diminishing confidence in official traceability (Andrade et al., 2023; Franca et al., 2023). To mitigate illegal logging, it is essential to implement an independent verification method grounded in material evidence, which incorporates scientific methods for origin attribution (Brancalion et al., 2018). Several analytical approaches have been developed to verify the geographic origin of timber across both temperate and tropical regions, including dendrochronological and anatomical analyses, multi-elemental chemistry, and genetic fingerprinting (Akhmetzyanov et al., 2019; Delmás et al., 2020; Hornink et al., 2025; Boeschoten et al., 2025). These methods provide complementary lines of evidence and can be combined to strengthen wood traceability frameworks.

Among them, stable isotopes serve as intrinsic markers of provenance because they can imprint distinct isotopic compositions in organic tissues growing at different locations. Spatial variations in temperature, water availability, altitude, soil prop-

erties, and lithology drive isotope variations in the environment, and those variations are propagated to plants and ecosystems (Leavitt and Long, 1984; Deleens et al., 1994; McCarroll and Loader, 2004; Fry, 2006; Leavitt and Roden, 2022).

In  $C_3$  plants, which include all tree species, carbon isotopes are assimilated from atmospheric  $CO_2$  that diffuses from the atmosphere to the leaf carboxylation sites, where the rubisco enzyme converts them into carbohydrates that are the source of the entire tree's carbon metabolism. The  $p_i/p_a$  ratio is influenced by both  $CO_2$  diffusion into the leaf and carboxylation rate and is directly linked to carbon isotope fractionation (Farquhar et al., 1982, 1989). Despite their identical photosynthetic pathways,  $C_3$  plants can exhibit a broad range of  $\delta^{13}$ C values, particularly in forested ecosystems, influenced by the "canopy effect" and by stomatal conductance (Farquhar et al., 1989; Medina and Minchin, 1980; Ometto et al., 2006). Plants growing in closed-canopy forests tend to show lower (more negative)  $\delta^{13}$ C values because of the incorporation of respired  $CO_2$ , which has lower  $CO_2$  diffusion (Farquhar et al., 1989; Martinelli et al., 2021). Regional  $CO_2$  values in forests usually reflect the interactions of these processes with primary productivity, canopy density, climate, and water availability, which usually play important roles (O'Leary, 1981; Farquhar et al., 1982; McCarroll and Loader, 2004; Fry, 2006). However, carbon isotopes have rarely been used as a marker of provenance because intra-site  $CO_2$  variations are large (Martinelli et al., 1998; Paredes-Villanueva et al., 2022). At a given site, co-occurring tropical forest tree species often show large  $CO_2$  differences, reflecting variations in successional status, canopy position, and consequently in water use efficiency (WUE) (Guehl et al., 1998; Bonal et al., 2000).

Nitrogen isotopes in plants are primarily controlled by the source of nitrogen assimilated because uptake of nitrogen by plants leads to negligible fractionation (Högberg, 1997). However, soil nitrogen cycling and soil microbial processes strongly fractionate nitrogen isotopes, and those  $\delta^{15} N$  variations are propagated to plants (Robinson, 2001; Craine et al., 2015). Among the key processes fractionating nitrogen isotopes, nitrification and the subsequent leaching of soluble nitrates out of soils in wet environments can leave soils with higher  $\delta^{15} N$  values (Mariotti et al., 1981; Craine et al., 2015). Denitrification through the diffusion of  $^{15}N$ -depleted gases also leaves soils with higher  $\delta^{15}N$  values (Mariotti et al., 1981; Wang et al., 2018). Ectomycorrhizal fungi also play a key role in controlling  $\delta^{15}N$  values in plants by influencing how tightly nitrogen cycles in soils, thereby limiting or amplifying fractionation processes (Craine et al., 2009). Nitrogen-fixing legumes, in principle, can also lower foliar  $\delta^{15}N$  values because atmospheric  $N_2$  has a  $\delta^{15}N$  close to 0% (Evans, 2001; Vitousek et al., 2002).

In tropical forests, however, soils and foliage are generally higher in  $^{15}$ N compared with other biomes, reflecting an open nitrogen cycle dominated by losses (Martinelli et al., 1999). In the Amazon, this pattern is especially evident; although nitrogen-fixing trees are present, in mature *terra firme* forests they often contribute little to the nitrogen balance because most legumes rely on soil nitrogen rather than atmospheric fixation (Ometto et al., 2006; Nardoto et al., 2008). Moreover, ectomycorrhizal associations are relatively rare, with arbuscular mycorrhizae predominating (Corrales et al., 2018). Since AM fungi exert weaker control on isotopic fractionation, Amazonian plants tend to reflect the  $\delta^{15}$ N values of the soil nitrogen pools more directly.

The  $\delta^{15}$ N of plants is not determined solely by the isotopic composition of soil nitrogen but is also influenced by internal physiological processes such as assimilation, redistribution, and resorption (Evans, 2001). Beyond this physiological control,

https://doi.org/10.5194/egusphere-2025-5452 Preprint. Discussion started: 13 November 2025

© Author(s) 2025. CC BY 4.0 License.

 $\delta^{15}$ N values in the Amazon are further shaped by edaphic properties and landscape position, which regulate the relative importance of processes such as nitrification, leaching, and denitrification (Martinelli et al., 1999; Amundson et al., 2003; Ometto et al., 2006; Nardoto et al., 2014).

Although both isotopes respond to fine-scale environmental and physiological conditions that may introduce some within-site heterogeneity, coherent large-scale patterns of  $\delta^{13}$ C and  $\delta^{15}$ N have been consistently observed across South America, particularly within the Amazon Basin (Martinelli et al., 2021). In the Brazilian Amazon, soil  $\delta^{15}$ N values are higher in the east than in the west (Sena-Souza et al., 2020), with lower isotopic compositions occurring in the nutrient-rich, younger soils of western Amazonia and higher values in the highly weathered, nutrient-poor soils of the eastern region (Martinelli et al., 1999; Nardoto et al., 2014). Across the continent, foliar  $\delta^{13}$ C values are lowest in tropical forests, particularly within the Amazon Basin (Powell et al., 2012). Within the basin,  $\delta^{13}$ C values become progressively less negative from the humid core areas toward the drier transition zones and forest peripheries (Ometto et al., 2002; Martinelli et al., 2007). This coherent spatial structuring highlights the strong environmental control on isotopic composition and underscores the potential of combining carbon and nitrogen isotopes as complementary intrinsic tracers for determining provenance within tropical ecosystems.

Isotopic maps, or isoscapes, render these spatial patterns more explicit and are extensively used in the fields of ecology, biogeochemistry, food authentication, and wildlife tracking (Bowen, 2010; Bataille and Bowen, 2012; Watkinson et al., 2020, 2022; Bataille et al., 2021; Reich et al., 2021; Le Corre et al., 2025). By providing reference maps, isoscapes facilitate data visualization and interpretation. In forensic contexts, they can provide indications or material evidence recognized in judicial processes (Ehleringer and Matheson Jr., 2010; Chesson et al., 2018; Meier-Augenstein, 2017). Among the various methods used to construct isoscapes, recent studies have increasingly employed machine-learning techniques to model complex and nonlinear isotopic patterns across landscapes (Barberena et al., 2021; Reich et al., 2024; Holt et al., 2025).

The absence of carbon and nitrogen isoscapes for Amazonian wood precludes the potential use of these isotopes for origin attribution methodologies, along with other isotopes (e.g. strontium and oxygen) or other provenance markers (e.g. genetics). We aim at testing whether  $\delta^{13}$ C and  $\delta^{15}$ N in wood across the Amazon are predictable and contain provenance information for Amazonian wood at the basin scale, and whether taxonomic effects are overridden by environmental and biogeochemical controls at this scale. We first generated a dataset of carbon and nitrogen isotopes in wood across the Amazon. We then leveraged machine-learning approaches to predict the spatial variability of carbon and nitrogen isotopes across the landscape.

## 2 Materials and methods

## 2.1 Study area and sampled species

From a representative collection of 571 trees encompassing 75 genera and 25 botanical families distributed across 47 sites throughout the Amazon, taxonomic identification was performed at the genus level because species-level classification is particularly challenging in tropical forests with high floristic diversity. A subset of samples originated from naturally fallen timber found in the field, which limited the possibility of taxonomic identification. These were classified as "unknown," representing approximately 5% of the total dataset (29 individuals).

The selection of species prioritized taxa with high commercial value in the Brazilian Amazon, specifically those identified as most exploited by IMAFLORA (Andrade et al., 2022). Due to logistical limitations in the sampling of certain target species, additional genera with lower economic significance were included based on availability through partner research groups. Nonetheless, the majority of the sampled individuals belong to genera acknowledged for their commercial importance in the region (Fig. S1).

The selection of the 47 sampling sites aimed at maximizing representation of the region's spatial and ecological variability, with sites encompassing a broad geographic distribution across the Brazilian Amazon (Fig. 1). In practice, while some samples were precisely georeferenced, the majority of the samples were collected within a site to simplify collection for our forestry partners. In general, sites were defined as zones of a few square kilometers where sampling of individual trees was conducted along the road within a few kilometers (min = 0 km, max = 215 km, median = 0.9 km).

## 2.2 Sample collection and preparation

The minimum diameter of sampled trees adhered to the legal harvesting threshold (DBH ≥ 50 cm) as stipulated by Brazilian regulations (IBAMA, 2006; CONAMA, 2009). To ensure the geographic traceability of the samples, the majority of sites were situated within protected areas, while the other sites comprised certified sustainable forest management areas and long-term forest research plots. Geographic coordinates of the sampling sites and individual trees are provided in Table S3, which is available in the OSF repository linked to this paper instead of being included directly in the Supplementary Information.

From each tree, a cross-sectional disk or wedge was excised at the basal part of the trunk, preserving the complete radial profile inclusive of the pith, heartwood, and sapwood. The disks measured approximately 5–6 cm in thickness. In the laboratory, each specimen was transversely sectioned into a strip with dimensions of 1.5 cm in width and 2.5 mm in thickness, and subsequently oven-dried at 70  $^{\circ}$ C for 24 hours, following standard protocols for tropical wood anatomical and densitometric preparation (Quintilhan et al., 2021). Subsequent sampling was guided by the work of Batista et al. (2025), who evaluated isotopic variation along the radial axis by excising five 1.5  $\times$  2 cm segments at distinct anatomical positions (pith, heartwood, heartwood–sapwood transition, and sapwood). Their results showed that the heartwood–sapwood transition provides a reliable approximation of mean isotopic composition across the entire radius. Based on this evidence, we targeted this specific segment for the present study (Fig. 2), as it combines analytical representativeness with practical advantages for forensic applications, since it is typically preserved in processed timber, whereas the sapwood is often removed during industrial processing.

# 140 **2.3 Isotope analyses**

130

135

For the purpose of conducting isotopic analysis, the samples were meticulously ground using a RETSCH ZM 300 mixer mill to ensure thorough homogenization. All procedures were executed within a climate-controlled environment to minimize the potential for contamination. The resultant powdered samples were subsequently weighed using an analytical balance, with masses spanning from 2.7 to 3.0 mg to ascertain the requisite quantity for analysis. Thereafter, the pulverized material was encapsulated within tin capsules, which are appropriate for the determination of  $^{13}$ C/ $^{12}$ C and  $^{15}$ N/ $^{14}$ N isotopic ratios.

**Figure 1.** Sampling sites selection. **(a)** South America with emphasis on the Brazilian Amazon region. **(b)** Location of the collected individuals across 47 sites in the Brazilian Amazon. Numbers correspond to site names listed in Table 1. The map highlights the states where the sampling took place, with grey lines indicating state boundaries, yellow points representing the sampled locations, and blue points (20 – Feijó and 24 – Itapuã do Oeste) corresponding to the sites analyzed in detail in this study. The South America boundaries are from (Natural Earth, 2023), and the Amazon biome polygon is from (Assis et al., 2019).

The isotopic analyses were performed utilizing a Carlo Erba 1110 elemental analyzer (Milan, Italy), coupled a isotope ratio mass spectrometer (Delta plus 2, Thermo Fisher Scientific, Bremen, Germany) for the determination of isotopic ratios. Analyses were undertaken at the Isotopic Ecology Laboratory of the Center for Nuclear Energy in Agriculture (*LEI/CENA/USP*, Brazil).

The isotopic ratio (R) was defined as the ratio of the rare to the abundant isotope in the sample, expressed relative to an international reference standard. Isotopic values were calibrated against the Vienna Pee Dee Belemnite (VPDB) and AIR scales for carbon and nitrogen, respectively. For carbon and nitrogen isotope analyses, internationally certified reference materials

**Figure 2.** Field sampling procedure. (1) Cross-sectional disk excised at the base of the trunk, preserving the anatomical structure of the wood. (2) Disk sectioned with approximately 5–6 cm thickness. (3) Transversal sectioning of the wood strip from the disk and subsequent sampling of the transition zone between heartwood and sapwood along the radial axis for isotope analysis.

NBS-22 ( $\delta^{13}C = -30.03 \pm 0.04 \%$  VPDB), IAEA-N-1 ( $\delta^{15}N = +0.43 \pm 0.07 \%$  AIR), and IAEA-N-2 ( $\delta^{15}N = +20.41 \pm 0.12 \%$  AIR) were employed for calibration. The internal laboratory standard consisted of a sugarcane material routinely analyzed as a blind quality-control sample. This internal standard was measured repeatedly throughout each analytical sequence to monitor instrument stability and analytical precision. Reported values remained within the expected range, with standard uncertainties of  $\pm 0.10 \%$  for  $\delta^{13}C$  and  $\pm 0.15 \%$  for  $\delta^{15}N$ .

Isotopic composition is expressed in delta ( $\delta$ ) notation, with values reported in per mil (%) relative to international reference standards (e.g., VPDB for carbon, AIR for nitrogen), following IUPAC recommendations (Coplen, 2011; Skrzypek et al., 2022), and defined as:

$$160 \quad \delta^{i/j}E = \frac{R_{\text{sample}}}{R_{\text{standard}}} - 1 \tag{1}$$

where R represents the ratio of the heavy (iE) to the light (jE) isotope of element E (e.g.,  $^{13}\text{C}/^{12}\text{C}$  or  $^{15}\text{N}/^{14}\text{N}$ ).

## 2.4 Intra-site variability

155

165

All data processing, statistical analyses, and figure generation were performed in R version 4.3.3 (R Core Team, 2025). While the goal of this study is to primarily assess the inter-site variability and the potential of using carbon and nitrogen isotopes for tracing wood origin, we first examined intra-site isotopic variance to evaluate the portion of the total variance that is not dependent on spatial autocorrelation. In this study, we sampled trees of different ages, species, heights, and rooting depths, which are known to influence carbon and nitrogen isotopes as shown by Batista et al. (2025). Additionally, the loose definition of "site" in this study, as a zone of several square kilometers potentially encompassing distinct geomorphology, geology, microclimates, and environments, would inherently lead to some intra-site variability.

For each site, we calculated the median, mean, standard deviation (SD), and root mean square error (RMSE) for  $\delta^{13}$ C and  $\delta^{15}$ N values. Boxplots were generated to illustrate the distribution of values within each site. These descriptive statistics provide the basis to evaluate the intra-site uncertainty for each isotope and the spatial resolution limitations of their respective predictive models.

To further disentangle geographic and taxonomic sources of variability, we conducted specific analyses on three genera (*Cedrela*, *Handroanthus*, and *Dipteryx*) that co-occurred in two well-sampled sites (*Feijó* and *Itapuã do Oeste*). Site effects were tested using independent-sample *t*-tests, while genus effects within sites were assessed using one-way ANOVA followed by pairwise comparisons with false discovery rate (FDR) correction. We also calculated 95% confidence intervals (CI) around mean values to quantify within-site dispersion at the genus level. These combined analyses allowed us to evaluate whether isotopic variability was more strongly influenced by taxonomic identity or geographic origin.

## 180 2.5 Inter-site variability and isoscape development

In order to predict  $\delta^{13}$ C and  $\delta^{15}$ N values in Amazonian wood, a multivariate regression approach was employed, following methodologies established in prior isoscape studies (e.g., Bataille et al., 2018, 2020; Sena-Souza et al., 2020; Reich et al., 2021; Le Corre et al., 2025; Martinelli et al., 2025). The process involved four main steps: (1) collection of auxiliary variables; (2) application of a Random Forest regression algorithm; (3) assessment of model performance through cross-validation and prediction error metrics; and (4) spatial application of the model to produce predictive isoscapes and corresponding uncertainty maps. The subsequent sections elaborate on each step in detail.

## 2.5.1 Auxiliary variables




To predict the spatial variability of  $\delta^{13}$ C and  $\delta^{15}$ N values within Amazonian wood, we assembled an extensive array of 74 spatially explicit covariates that encompass critical environmental, climatic, edaphic, geophysical, and biological factors (Table S1). These variables were selected based on their established direct or indirect associations with plant isotopic composition and their ability to represent processes influencing carbon and nitrogen assimilation within tropical forest ecosystems.

The dataset includes soil physicochemical properties such as pH, cation exchange capacity, clay content, bulk density, total nitrogen, and organic carbon stocks, which govern nitrogen mineralization, microbial activity, and root nutrient uptake factors intrinsically linked to  $\delta^{15}$ N variation (Amundson et al., 2003; Nardoto et al., 2008; Craine et al., 2015; Savard and Daux, 2020; Brunello et al., 2024). Climatic variables, including temperature, precipitation, vapor pressure, aridity index, and potential evapotranspiration, affect stomatal conductance and photosynthetic discrimination, which are primary drivers of  $\delta^{13}$ C variability in C<sub>3</sub> plants (Farquhar et al., 1982; Locosselli et al., 2013; Van Der Sleen et al., 2014; Martinelli et al., 2021).

We also incorporated remotely sensed indicators of vegetation structure and function, such as canopy greenness, leaf area, net primary productivity, and carbon assimilation. These proxies reflect photosynthetic capacity and ecosystem productivity, consequently impacting plant nutrient demand and isotopic signatures (Ometto et al., 2006; Cernusak et al., 2013). Wood density, an ecophysiological trait related to water-use efficiency, was included as a proxy for genus-specific physiological strategies (Halder et al., 2024).






Furthermore, we examined topographic attributes such as elevation, geographic gradients like proximity to the coastline, and an array of geological and geophysical variables encompassing bedrock age, porosity, and the atmospheric deposition of dust and sea salt. A selection of raster layers utilized in this investigation was acquired in a pre-processed format from a geospatial database assembled by Bataille et al. (2018, 2020, 2021), originally designed for global and regional isoscape modeling of strontium isotopes. Additional covariates were independently acquired and processed from publicly accessible datasets to ensure comprehensive spatial coverage and thematic complementarity. All environmental geospatial products were resampled and reprojected to the equivalent WGS84–Eckert IV projection, with a spatial resolution of 1 km<sup>2</sup>, to standardize the raster layers and mitigate area distortions at a global scale.

## 2.5.2 Random forest regression and geospatial predictions

Random Forest constitutes a tree-based machine learning algorithm that constructs multiple decision trees by employing bootstrap sampling alongside random feature selection (Breiman, 2001). It synthesizes their outputs to estimate a response variable while obviating the necessity for data transformation or presumptions regarding distribution or residual variance (Bataille et al., 2020). In accordance with the general Random Forest regression methodology established by Bataille et al. (2018) for  ${}^{87}\text{Sr}/{}^{86}\text{Sr}$  isoscapes and implemented using the caret package (Kuhn, 2008), we refined and applied this approach to model the spatial distribution of  $\delta^{13}\text{C}$  and  $\delta^{15}\text{N}$ .

Before the initiation of model training, environmental variables were extracted from raster layers corresponding to each individual sampling point and submitted in full to the VSURF (Variable Selection Using Random Forest) algorithm. After the selection process, a Pearson correlation analysis was performed among the selected variables to identify highly correlated pairs (r > 0.90), which were considered redundant. Additionally, some variables with lower predictive contribution were excluded to improve model parsimony and reduce error. Although Random Forest algorithms are inherently robust to multicollinearity, these post-selection refinements aimed to enhance both the ecological interpretability and the predictive performance of the final model.

Discrepancies in spatial georeferencing were noted among the sampling sites; in some instances, individual trees were assigned unique coordinates, while in other cases, all individuals from a single site shared a common coordinate. To mitigate spatial pseudoreplication, data were aggregated at the site level. For each site, median values of latitude, longitude,  $\delta^{13}$ C, and  $\delta^{15}$ N were computed, resulting in 47 unique site-level entries.

Subsequently, site-level entries were utilized to derive the values of 74 environmental variables from raster datasets, culminating in a comprehensive regression matrix. To discern the most pertinent predictors of  $\delta^{13}$ C and  $\delta^{15}$ N, the VSURF algorithm (Genuer et al., 2015) was employed, executing 3,000 trees at each phase. Variable selection was predicated on the "prediction" phase of the algorithm, which isolates the subset of predictors that most effectively reduce prediction error.

The designated predictors were subsequently employed to construct Random Forest regression models using the caret package, which incorporated hyperparameter tuning and employed repeated 10-fold cross-validation (with five repetitions), with 80% of the data allocated for training during each iteration. Model efficacy was assessed using the coefficient of determination ( $R^2$ ), root mean square error (RMSE), and mean absolute error (MAE).






To elucidate the contribution of each variable to the model predictions, we calculated variable importance metrics grounded in the framework of the Random Forest algorithm. Subsequently, for all selected covariates, partial dependence plots (PDPs) were generated to illustrate the marginal effects of each geospatial predictor on the isotopic ratios of  $\delta^{13}$ C and  $\delta^{15}$ N (Friedman, 2001). Additionally, to investigate potential interaction effects between environmental variables on isotopic values, bivariate PDPs were constructed using the two most influential covariates. These plots facilitate a more comprehensive interpretation of the combined influence of predictors on the predicted isotopic values, thereby enhancing the ecological understanding of the observed spatial patterns.

Utilizing the refined Random Forest model and the selected predictors, isoscapes were developed to depict the mean predicted  $\delta^{13}$ C and  $\delta^{15}$ N values in wood across the Amazon region. To ensure spatial consistency with regional datasets, the final maps were reprojected to the SIRGAS 2000 geographic coordinate system (EPSG:4674), which is suitable for South America. To appraise the spatial uncertainty associated with these predictions, a crucial process in evaluating the reliability of isoscapes, we employed the Quantile Regression Forest (QRF) model, as delineated by Meinshausen (2006) and Le Corre et al. (2025). This algorithm facilitates the estimation of prediction intervals based on the distribution of outputs from the ensemble of decision trees. For each pixel, the 15.9th and 84.1st quantiles were extracted, roughly corresponding to  $\pm 1$  standard deviation under the assumption of normally distributed residuals. Due to computational memory constraints, predictions were performed in successive blocks of 100,000 pixels containing complete environmental data. Spatial uncertainty was subsequently computed as half the difference between the upper and lower quantiles, resulting in a continuous raster that represents the standard error of the prediction for each pixel. This methodology enabled the development of high-resolution uncertainty maps (1 km²), which can be integrated into probabilistic geographic assignment models, thereby augmenting the robustness and reliability of forensic and ecological applications based on  $\delta^{13}$ C and  $\delta^{15}$ N isotopes.

#### 3 Results

## 3.1 Inter- and intra-site variability

The isotopic composition of all 571 wood samples adhered to a normal distribution for both  $\delta^{13}$ C and  $\delta^{15}$ N (Fig. S2). The mean  $\delta^{13}$ C value was  $-28.1 \pm 1.4$  %, with a 95% confidence interval extending from -28.2 % to -28.0 %. The  $\delta^{13}$ C values encompassed a total range of 9.7 %, varying from a minimum of -33.7 % to a maximum of -24.0 %. Conversely,  $\delta^{15}$ N values averaged  $4.0 \pm 2.3$  %, with a 95% confidence interval of 3.8 % to 4.2 %, and fluctuated from -2.7 % to 11.2 %, resulting in a total amplitude of 13.9 %.

Site-level variability in  $\delta^{13}$ C (Fig. 3a) and  $\delta^{15}$ N (Fig. 3b) is summarized in Table 1 and visualized through boxplots, which emphasize both within- and between-site variation. Concerning  $\delta^{13}$ C, mean site values were lowest in  $Juru\acute{a}$  ( $-30.5 \pm 1.4 \%$ ) and Atalaia do Norte ( $-30.4 \pm 1.1 \%$ ), both located in the western Amazon, and highest in  $Rur\acute{o}polis$  ( $-26.7 \pm 1.0 \%$ ) and Ferreira Gomes ( $-26.6 \pm 0.9 \%$ ), situated along the eastern and northeastern edges of the basin. The highest intra-site variability was observed in Alvorada d'Oeste (SD  $\pm 1.9 \%$ ); RMSE = 1.63 %); n = 4),  $Uruar\acute{a}$  (SD  $\pm 1.7 \%$ ); RMSE = 1.53 %); n = 6), and Pauini (SD  $\pm 1.5 \%$ ); RMSE = 1.50 %); n = 41), whereas the lowest variability occurred in Porto  $Esperidi\~{a}o$ 

(SD  $\pm 0.3$  %c; RMSE = 0.28 %c; n = 5), Santa Maria das Barreiras (SD  $\pm 0.4$  %c; RMSE = 0.35 %c; n = 9), and Cáceres (SD  $\pm 0.4$  %c; RMSE = 0.38 %c; n = 12).

Table 1: Site-level summary of  $\delta^{13}$ C and  $\delta^{15}$ N in wood samples. Includes site number (n), site name, state, geographic coordinates, number of individuals, and the median, mean  $\pm$  SD, and RMSE for both isotopes.

|    | Site                   | State       | Lat.     | Long.    | N      | $\delta^{13}\mathrm{C}$ | $\delta^{13}{ m C}$    | $\delta^{13}\mathrm{C}$ | $\delta^{15} { m N}$ | $\delta^{15} { m N}$ | $\delta^{15} { m N}$ |
|----|------------------------|-------------|----------|----------|--------|-------------------------|------------------------|-------------------------|----------------------|----------------------|----------------------|
| _  |                        |             |          |          | indiv. | Median                  | Mean $\pm$ SD          | RMSE                    | Median               | Mean $\pm$ SD        | RMSE                 |
| 1  | Almeirim               | Pará        | -0.9737  | -53.3244 | 10     | -29.1                   | $-29.2 \pm 0.6$        | 0.57                    | 2.5                  | $2.7\pm1.5$          | 1.4                  |
| 2  | Altamira               | Pará        | -3.312   | -52.0548 | 1      | -28.5                   | -28.5 $\pm$ NA         | 0                       | 4.7                  | $4.7 \pm \text{NA}$  | 0                    |
| 3  | Alvorada D'Oeste       | Rondônia    | -11.2231 | -62.1674 | 4      | -28.8                   | $-29 \pm 1.9$          | 1.63                    | 4.2                  | $4.2 \pm 0.5$        | 0.37                 |
| 4  | Ariquemes              | Rondônia    | -9.8544  | -62.7483 | 24     | -27.7                   | $-27.6 \pm 1.1$        | 1.12                    | 6.3                  | $6.1\pm2$            | 1.94                 |
| 5  | Atalaia do Norte       | Amazonas    | -4.3037  | -70.2911 | 9      | -30.3                   | $-30.4 \pm 1.1$        | 0.99                    | 2.6                  | $2.3\pm1.1$          | 1.05                 |
| 6  | Barcelos               | Amazonas    | -1.764   | -62.382  | 18     | -29.3                   | $-29.3 \pm 1.6$        | 1.57                    | 3.9                  | $3.6 \pm 1.7$        | 1.63                 |
| 7  | Barra do Bugres        | Mato Grosso | -14.9064 | -57.9927 | 9      | -28.8                   | $\text{-}28.7 \pm 0.6$ | 0.55                    | -                    | -                    | -                    |
| 8  | Belterra               | Pará        | -3.4961  | -54.9659 | 33     | -27.5                   | $-27.7 \pm 1$          | 1.02                    | 5                    | $5\pm1.9$            | 1.92                 |
| 9  | Buritis                | Rondônia    | -10.0969 | -63.8464 | 9      | -28.6                   | $-28.5 \pm 0.9$        | 0.87                    | 6.1                  | $6.2\pm0.8$          | 0.7                  |
| 10 | Cáceres                | Mato Grosso | -16.2149 | -58.364  | 12     | -28.2                   | $-28.1 \pm 0.4$        | 0.38                    | 5.6                  | $5.6 \pm 1.2$        | 0.88                 |
| 11 | Cacoal                 | Rondônia    | -11.3714 | -61.3322 | 3      | -27.8                   | $-28.2 \pm 0.6$        | 0.51                    | 5.3                  | $5.3\pm1.5$          | 1.05                 |
| 12 | Campo Novo de Rondônia | Rondônia    | -10.3249 | -63.6574 | 1      | -28.3                   | -28.3 $\pm$ NA         | 0                       | 2.5                  | $2.5 \pm \text{NA}$  | 0                    |
| 13 | Candeias do Jamari     | Rondônia    | -8.6258  | -62.8445 | 21     | -27.5                   | $-27.8 \pm 1.1$        | 1.09                    | 7.3                  | $7.5\pm1.9$          | 1.82                 |
| 14 | Caracaraí              | Roraima     | 1.6003   | -61.1219 | 40     | -27.6                   | $-27.8 \pm 1.3$        | 1.24                    | 2.2                  | $2.3 \pm 2.5$        | 2.47                 |
| 15 | Castanheiras           | Rondônia    | -11.4167 | -61.76   | 3      | -29.5                   | $-29.1 \pm 0.7$        | 0.58                    | 2.6                  | $2.6 \pm 0.5$        | 0.36                 |
| 16 | Colniza                | Mato Grosso | -9.3659  | -59.0516 | 8      | -27.3                   | $-27.2 \pm 0.8$        | 0.74                    | 4.4                  | $4.3\pm1.7$          | 1.6                  |
| 17 | Comodoro               | Mato Grosso | -13.23   | -60.2674 | 4      | -27.4                   | $-27.2 \pm 0.6$        | 0.54                    | 1.1                  | $0.6 \pm 2.5$        | 2.13                 |
| 18 | Cotriguaçu             | Mato Grosso | -9.355   | -58.8657 | 5      | -28                     | $-28.1 \pm 0.6$        | 0.5                     | 4.6                  | $4.9 \pm 1.8$        | 1.58                 |
| 19 | Espigão D'Oeste        | Rondônia    | -11.5406 | -61.0067 | 10     | -28                     | $-28.2 \pm 1.5$        | 1.43                    | 3.3                  | $3.7 \pm 1.6$        | 1.48                 |
| 20 | Feijó                  | Acre        | -8.6447  | -70.1453 | 30     | -27.9                   | $-27.7 \pm 1.3$        | 1.3                     | 1.1                  | $1.6\pm1.6$          | 1.53                 |
| 21 | Ferreira Gomes         | Amapá       | 1.0918   | -51.937  | 27     | -26.6                   | $-26.6 \pm 0.9$        | 0.86                    | 4.7                  | $4.5\pm1.4$          | 1.36                 |
| 22 | Itaituba               | Pará        | -5.9788  | -55.2672 | 10     | -29.5                   | $-29.4 \pm 0.7$        | 0.64                    | 4.1                  | $4.1\pm1.1$          | 1.06                 |
| 23 | Itapiranga             | Amazonas    | -2.495   | -59.1215 | 30     | -27.6                   | $-27.5 \pm 0.9$        | 0.93                    | 3.1                  | $2.8\pm1.7$          | 1.66                 |
| 24 | Itapuã do Oeste        | Rondônia    | -9.363   | -63.0818 | 50     | -27.9                   | $-27.7 \pm 1$          | 0.97                    | 5.4                  | $5.5\pm1.3$          | 1.29                 |
| 25 | Jaru                   | Rondônia    | -10.3988 | -62.5056 | 1      | -28.2                   | -28.2 $\pm$ NA         | 0                       | 2.6                  | $2.6 \pm \text{NA}$  | 0                    |

| n  | Site                      | State       | Lat.     | Long.    | N<br>in div | δ <sup>13</sup> C | δ <sup>13</sup> C        | $\delta^{13}\mathrm{C}$ | $\delta^{15} \mathrm{N}$ | $\delta^{15}  m N$<br>Mean $\pm$ SD | $\delta^{15}N$ |
|----|---------------------------|-------------|----------|----------|-------------|-------------------|--------------------------|-------------------------|--------------------------|-------------------------------------|----------------|
|    |                           |             |          |          | maiv.       | Median            | Mean ± SD                | KMSE                    | Median                   | Mean ± SD                           |                |
| 26 | Ji-Paraná                 | Rondônia    | -10.8567 | -61.9504 | 6           | -28.6             | $\textbf{-28.7} \pm 0.5$ | 0.44                    | 2.4                      | $2.8\pm1.2$                         | 1.11           |
| 27 | Juruá                     | Amazonas    | -3.9072  | -66.0551 | 6           | -30.3             | $\text{-}30.5 \pm 1.4$   | 1.31                    | 3.8                      | $4\pm0.6$                           | 0.56           |
| 28 | Manicoré                  | Amazonas    | -6.0097  | -61.8687 | 6           | -29.8             | $-30.1 \pm 1.3$          | 1.15                    | 3.8                      | $3.6 \pm 1.3$                       | 1.17           |
| 29 | Maués                     | Amazonas    | -3.9955  | -57.5893 | 5           | -29.5             | $-29.1 \pm 1$            | 0.88                    | 3.6                      | $3.6 \pm 0.8$                       | 0.72           |
| 30 | Monte Alegre              | Pará        | -0.9907  | -54.185  | 11          | -28.4             | $-28.4 \pm 0.9$          | 0.85                    | 4.5                      | $4.2\pm1.2$                         | 1.16           |
| 31 | Monte Negro               | Rondônia    | -10.0734 | -63.2122 | 1           | -27.9             | $-27.9 \pm NA$           | 0                       | 9                        | $9 \pm \text{NA}$                   | 0              |
| 32 | Nova Brasilândia D'Oeste  | Rondônia    | -11.7336 | -62.3853 | 6           | -27.5             | $-28 \pm 1$              | 0.89                    | 5                        | $5.9 \pm 2.8$                       | 2.48           |
| 33 | Pauini                    | Amazonas    | -8.4844  | -69.0357 | 41          | -27.6             | -27.5 ± 1.5              | 1.5                     | 2.4                      | $2.6 \pm 1.3$                       | 1.33           |
| 34 | Pimenta Bueno             | Rondônia    | -11.605  | -61.1153 | 3           | -28.3             | $-28.5 \pm 0.6$          | 0.51                    | 3.2                      | $3.2 \pm 0$                         | 0.01           |
| 35 | Porto Esperidião          | Mato Grosso | -15.7962 | -58.54   | 5           | -28.7             | $-28.7 \pm 0.3$          | 0.28                    | -                        | -                                   | -              |
| 36 | Porto Velho               | Rondônia    | -9.5426  | -64.1413 | 12          | -28.4             | $-28.3 \pm 1.3$          | 1.28                    | 6.8                      | $6.3\pm2.9$                         | 2.71           |
| 37 | Primavera de Rondônia     | Rondônia    | -11.8458 | -61.2819 | 13          | -29.1             | $-28.7 \pm 1.2$          | 1.19                    | 3.4                      | $4\pm1.7$                           | 1.58           |
| 38 | Rolim de Moura            | Rondônia    | -11.6563 | -61.6744 | 2           | -28               | $-28 \pm 0.7$            | 0.5                     | 2.5                      | $2.5 \pm \text{NA}$                 | 0              |
| 39 | Rorainópolis              | Roraima     | 0.9036   | -60.4341 | 9           | -27               | $-27.3 \pm 0.7$          | 0.67                    | 6                        | $5.8 \pm 1.5$                       | 1.44           |
| 40 | Rurópolis                 | Pará        | -3.9923  | -54.908  | 11          | -26.4             | $-26.7 \pm 1$            | 0.94                    | 4.3                      | $4.7\pm1.4$                         | 1.31           |
| 41 | Santa Maria das Barreiras | Pará        | -8.7159  | -50.4311 | 9           | -28.7             | $-28.8 \pm 0.4$          | 0.35                    | -                        | -                                   | -              |
| 42 | São Gabriel da Cachoeira  | Amazonas    | -0.1212  | -67.0131 | 11          | -29.4             | $-29.7 \pm 1.2$          | 1.18                    | 3.5                      | $3.5 \pm 0.9$                       | 0.9            |
| 43 | Tanguro                   | Mato Grosso | -13.079  | -52.3864 | 9           | -28.1             | $-28 \pm 1.2$            | 1.17                    | 2.5                      | $2.2\pm1$                           | 0.99           |
| 44 | Theobroma                 | Rondônia    | -10.2654 | -62.3743 | 1           | -27.7             | $-27.7 \pm NA$           | 0                       | 7.1                      | $7.1 \pm NA$                        | 0              |
| 45 | Uruará                    | Pará        | -2.9735  | -53.871  | 6           | -27.6             | $-28.2 \pm 1.7$          | 1.53                    | 4.5                      | $4.6\pm1.2$                         | 1.06           |
| 46 | Vitória do Xingu          | Pará        | -3.2719  | -51.8098 | 9           | -28.3             | $-28.3 \pm 0.7$          | 0.65                    | 5.6                      | $4.9 \pm 3$                         | 2.86           |
| 47 | ZF2                       | Amazonas    | -2.6391  | -60.1575 | 17          | -29.3             | -29.4 ± 1.5              | 1.44                    | 3.3                      | $3.4\pm1.9$                         | 1.82           |

Note: "-" indicates sites without isotopic data; "NA" marks unavailable statistics due to single observations.

Regarding  $\delta^{15}$ N, site mean values varied widely, with the lowest means observed in *Comodoro* (0.6  $\pm$  2.5 %) and *Feijó* (1.6  $\pm$  1.6 %), and the highest in *Candeias do Jamari* (7.5  $\pm$  1.9 %). The highest individual value (9.0 %) was recorded in *Monte Negro*, based on a single individual. The sites with the greatest intra-site dispersion were *Vitória do Xingu* (SD  $\pm$ 3.0 %); RMSE = 2.86 %; n = 9), *Porto Velho* (SD  $\pm$ 2.9 %; RMSE = 2.71 %; n = 12), and *Nova Brasilândia d'Oeste* (SD  $\pm$ 2.8 %; RMSE = 2.48 %; n = 6). In contrast, minimal  $\delta^{15}$ N variability was observed in *Pimenta Bueno* (SD  $\pm$ 0.0 %; RMSE = 0.01 %; n = 3), *Castanheiras* (SD  $\pm$ 0.5 %; RMSE = 0.36 %; n = 3), and *Alvorada d'Oeste* (SD  $\pm$ 0.5 %; RMSE = 0.37 %; n = 4).

Figure 3. Site-level distributions of  $\delta^{13}$ C and  $\delta^{15}$ N in wood, ordered by site median. Numbers along the y-axis correspond to site names listed in Table 1. Gray dots show individual trees; vertical lines indicate medians; boxes show interquartile ranges (IQR); whiskers extend to  $1.5 \times$  IQR. Sites were grouped by (a) mean annual relative humidity ( $\geq 70\%$  vs. 

To further evaluate intra- and inter-generic variability, we focused on three genera (Cedrela, Handroanthus, and Dipteryx) that co-occur in two well-sampled sites,  $Feij\acute{o}$  and  $Itapu\~a$  do Oeste. For  $\delta^{13}$ C, differences among genera were more pronounced than differences between sites ( $Feij\acute{o}-27.76\pm1.54~\%c$ ;  $Itapu\~a$  do  $Oeste-27.76\pm1.24~\%c$ ; (Fig. 4a), with no significant site effect (t=0.016, df=29, p=0.987). In  $Feij\acute{o}$ , all three genera differed significantly (p<0.02), whereas in  $Itapu\~a$  do Oeste only Cedrela and Handroanthus contrasted (p=0.0114). Handroanthus consistently showed the highest values ( $Feij\acute{o}$ :  $-26.3\pm0.95~\%c$ ;  $Itapu\~a$ :  $-27.0\pm0.46~\%c$ ), Cedrela the lowest ( $Feij\acute{o}$ :  $-29.6\pm0.86~\%c$ ;  $Itapu\~a$ :  $-28.9\pm0.92~\%c$ ), and Dipteryx intermediate ( $Feij\acute{o}$ :  $-27.7\pm0.76~\%c$ ;  $Itapu\~a$ :  $-27.1\pm1.20~\%c$ ). Within-site dispersion at the genus level was substantial (Table S2).

In contrast, for  $\delta^{15}$ N the site effect was more pronounced than the taxonomic effect (Fig. 4b). Individuals from *Itapuã do* 295 *Oeste*  $(5.67 \pm 1.19 \%)$  were systematically higher in  $^{15}$ N compared to those from  $Feij\acute{o}$   $(2.25 \pm 1.67 \%)$ , a difference that was highly significant (t = -6.30, df = 29, p < 0.0001). At the genus level, *Handroanthus* averaged  $3.88 \pm 1.09 \%$  in  $Feij\acute{o}$  versus  $6.13 \pm 0.88 \%$  in  $Itapu\~a$  do Oeste (p = 0.0049), Cedrela  $0.8 \pm 0.51 \%$  versus  $5.28 \pm 1.66 \%$  (p = 0.0004), and Dipteryx  $1.89 \pm 1.46 \%$  versus  $5.53 \pm 0.71 \%$  (p = 0.0039). In  $Feij\acute{o}$ , Handroanthus differed significantly from Cedrela (p = 0.001) and Dipteryx (p = 0.019), whereas in  $Itapu\~a$  do Oeste no inter-generic differences were detected (all p > 0.5). Within-site dispersion at the genus level was more variable (Table S2).

## 3.2 Random Forest regression and geospatial predictions

## 3.2.1 $\delta^{13}$ C isoscape


After variable selection using the VSURF algorithm, two climate-related covariates associated with atmospheric moisture were identified as the main predictors of  $\delta^{13}$ C in wood: mean annual vapor pressure (*r.vapor*) and mean annual relative humidity (*r.hurs\_mean*), both showing similar importance in node purity within the model. Following *n*-fold cross-validation (see Materials and Methods, Sect. 2.5.2), the Random Forest model explained 60% of the variance in observed  $\delta^{13}$ C values, with an RMSE of 0.59 ‰ and a MAE of 0.48 ‰.

The scatter plot of observed versus predicted  $\delta^{13}$ C values reveals an approximately linear relationship (Fig. 5a). However, the fitted regression line (black) deviates from the identity line (dashed), indicating a tendency to overestimate  $\delta^{13}$ C at the lower end of the distribution (more negative  $\delta^{13}$ C values) and to underestimate at the upper end. The highest density of points occurs within the intermediate  $\delta^{13}$ C range (-28 to -27 ‰), with fewer observations toward both extremes.

Spatially, residuals are heterogeneously distributed, with both positive and negative deviations scattered across different regions of the Amazon (Fig. S3). This lack of a clear geographic pattern suggests that the model does not exhibit systematic spatial bias and performs well at a regional scale, although some local environmental factors not captured by the predictor variables may still partially influence the observed variation in wood  $\delta^{13}$ C.

Partial dependence plots (PDPs) indicate nonlinear patterns between predicted  $\delta^{13}$ C values in wood and the two variables selected by the model: mean annual relative humidity (*r.hurs\_mean*) and mean annual vapor pressure (*r.vapor*) (Fig. 5b). For relative humidity,  $\delta^{13}$ C values remain relatively stable below 63%, increase between approximately 64% and 68%, and



Figure 4.  $\delta^{13}$ C and  $\delta^{15}$ N values in wood by genus (*Handroanthus*, *Cedrela*, and *Dipteryx*) across two sites (*Feijó* and *Itapuã do Oeste*). (a)  $\delta^{13}$ C and (b)  $\delta^{15}$ N distributions. Gray dots represent individual trees; vertical bars indicate medians; boxes represent interquartile ranges (IQR); whiskers extend to  $1.5 \times$  IQR. Different lowercase letters indicate significant differences among genera within the same site according to Tukey's HSD test (p 

Figure 5. Random Forest regression performance and climatic partial dependence plots (PDPs) for  $\delta^{13}$ C in wood. (a) Relationship between observed and predicted  $\delta^{13}$ C values, with a fitted regression line (black) and the 1:1 reference line (dotted). (b) PDPs of mean annual relative humidity (left) and mean annual vapor pressure (right) on predicted  $\delta^{13}$ C. Solid black lines represent the average predicted response when varying each predictor independently. Histograms below each plot indicate the distribution of observed values for the corresponding variable, providing context on sampling density. (c) Interaction surface between relative humidity and vapor pressure showing their combined influence on predicted  $\delta^{13}$ C. The color gradient represents predicted  $\delta^{13}$ C values (‰), with warmer tones indicating higher values and cooler tones indicating lower values. Black contour lines delineate zones of similar predicted values.






Based on the final model trained with 47 sampled sites, a predictive isoscape of  $\delta^{13}$ C in wood was generated for the entire Amazon region (Fig. 6a). Predicted values span from -32% to -25%, with lower  $\delta^{13}$ C values concentrated across much of the state of Amazonas and extending eastward along the Amazon River toward the northeastern portion of the basin. Conversely, higher values are predominantly found in the southern and southeastern sectors, as well as in parts of the central-eastern Amazon. This spatial structure likely reflects climatic gradients, particularly those driven by atmospheric vapor pressure and mean relative humidity.

The predictive uncertainty map, expressed as the standard deviation of model predictions (Fig. 6b), reveals greater uncertainty in areas with sparse sampling, particularly in the eastern, central, and northwestern portions of the Amazon. In contrast, regions with denser sampling coverage, especially in the southern part of the basin, show lower predictive uncertainty.

## 3.2.2 $\delta^{15}$ N isoscape

For  $\delta^{15}$ N, VSURF initially selected total soil nitrogen (*r.nitrogen*), soil organic carbon (*r.soc*), mean annual net primary productivity (*r.npp*), fire frequency (*r.fire*), atmospheric deposition of volcanic material (*r.volc*), and three geological age metrics (*r.minage\_geol*, *r.meanage\_geol*, and *r.maxage\_geol*). After identifying redundancies between predictors, only *r.meanage\_geol* was retained in the final model, as it was highly correlated with the other geological age metrics (r > 0.98). Likewise, only *r.soc* was selected, while *r.nitrogen* was eliminated due to high collinearity between the two variables. Both *r.fire* and *r.volc* provided negligible predictive power and were also removed to improve model parsimony and facilitate ecological interpretation.

The final model used three predictors: mean geological age, soil organic carbon, and mean annual net primary productivity to predict  $\delta^{15}$ N values. Among these, *r.meanage\_geol* contributed most significantly to node purity, followed by *r.soc* and *r.npp*. The Random Forest model accounted for 67% of the variance in the observed  $\delta^{15}$ N values, with a root mean square error (RMSE) of 1.10 % and a mean absolute error (MAE) of 0.93 %, as determined by cross-validation using 44 samples. Isotope measurements were unavailable for three additional sites due to insufficient nitrogen signal detection during analysis.

The scatter plot of observed versus predicted  $\delta^{15}N$  values demonstrates a predominantly linear trend across the entire range of variation (Fig. 7a). A systematic deviation is apparent, characterized by a tendency to underestimate elevated  $\delta^{15}N$  values and overestimate reduced ones. The majority of data points are densely populated in the intermediate sector, specifically between 3.5 ‰ and 5.5 ‰, whereas fewer observations are located at the extremes of the distribution.

Spatially, the residuals are distributed heterogeneously across the Amazon, with both positive and negative deviations dispersed among the sampling sites (Fig. S5). As observed for  $\delta^{13}$ C, this heterogeneous spatial distribution implies the absence of localized bias, thereby supporting the robustness of the  $\delta^{15}$ N model on a regional scale. Partial dependence plots (PDPs) predominantly indicate nonlinear associations between predicted  $\delta^{15}$ N values and the three selected predictors (Fig. 7b). Regarding mean geological age,  $\delta^{15}$ N values exhibit a near-linear increase until approximately 1000 Ma, remain constant between 1000–2000 Ma, and slightly decrease thereafter. Concerning soil organic carbon content,  $\delta^{15}$ N values gradually increase up to approximately 250 g kg<sup>-1</sup>, followed by a marked decline at higher concentrations. Net primary productivity exhibits a positive effect up to about 2100 gC m<sup>-2</sup> yr<sup>-1</sup>, after which stabilization or a slight decline is observed at increased productivity levels.

**Figure 6.**  $\delta^{13}$ C isoscape and associated uncertainty. (a) Predictive isoscape of  $\delta^{13}$ C in Amazonian wood samples across 47 sites (circles), showing spatial variation in predicted values (‰) based on the final Random Forest model. (b) Map of associated predictive uncertainty expressed as the standard deviation (SD) of predicted  $\delta^{13}$ C values. Darker red areas indicate regions with higher model uncertainty.

The interaction plot between mean geological age and soil organic carbon (SOC) (Fig. 7c) indicates that the highest predicted  $\delta^{15}$ N values ( $\sim$ 6 ‰) occur under the combined conditions of very old substrates (>1500 Ma) and relatively low SOC (150–200 g kg<sup>-1</sup>). In contrast, the lowest  $\delta^{15}$ N values ( $\sim$ 3 ‰) are found in younger soils (<1000 Ma) with higher SOC (>250 g kg<sup>-1</sup>). This gradient suggests that  $\delta^{15}$ N increases with substrate age when coupled with declining organic carbon content, highlighting that the joint effect of geology and SOC explains isotopic variability more robustly than either predictor

Figure 7. Random Forest regression performance and environmental partial dependence plots (PDPs) for  $\delta^{15}N$  in wood. (a) Relationship between observed and predicted  $\delta^{15}N$  values, with a fitted regression line (black) and the 1:1 reference line (dotted). (b) PDPs of mean geological age of the bedrock (left), soil organic carbon content (center), and mean annual net primary productivity (right) on predicted  $\delta^{15}N$  values. Solid black lines represent the average predicted response when varying each predictor independently. Histograms below each plot indicate the distribution of observed values for the corresponding variable, providing context on sampling density. (c) Interaction surface between mean geological age and soil organic carbon content showing their combined effect on predicted  $\delta^{15}N$ . The color gradient represents predicted  $\delta^{15}N$  values (‰), with warmer colors indicating higher values and cooler colors indicating lower values. Black contour lines delineate zones of similar predicted  $\delta^{15}N$  values.

alone. The mean geological age, soil organic carbon content, and net primary productivity layers identified as the model's main predictors are included in the Supplement for reference and interpretation of the predictive patterns (Fig. S6).

Utilizing the final model calibrated with 44 sampled sites, a predictive isoscape for  $\delta^{15}N$  isotopic composition in wood across the entire Amazon basin was developed (Fig. 8a). Estimated  $\delta^{15}N$  values span from approximately 2 ‰ to 9 ‰, with  $\delta^{15}N$  enrichment observed in the northern, southern, and central-eastern regions. Conversely, reduced values are predominantly observed in the other regions, with substantial depletion evident across extensive areas of the southwestern, southern, and eastern Amazon. This spatial distribution is primarily indicative of the influence of mean geological age, with supplementary effects from soil organic carbon content.

The predictive uncertainty map (Fig. 8b), represented as the standard deviation of  $\delta^{15}N$  predictions, indicates heightened uncertainty in areas characterized by sparse sampling, notably within the northern and eastern portions of the basin. Notably, substantial uncertainty was also discerned in the southern regions despite the presence of relatively dense sampling. This phenomenon may indicate local-scale variations in  $\delta^{15}N$ , limitations in the explanatory capacity of the chosen predictors, or unaccounted-for environmental processes impacting nitrogen dynamics.

#### 4 Discussion






This study provides the first  $\delta^{13}$ C and  $\delta^{15}$ N isoscapes for Amazonian wood, offering a spatially explicit framework to unravel the ecological and environmental factors shaping isotopic variability across the basin. Our results indicate that  $\delta^{13}$ C patterns are predominantly controlled by relative humidity and vapor pressure, reflecting climatic and physiological regulation of wateruse efficiency, whereas  $\delta^{15}$ N distributions are primarily influenced by geological age and soil organic carbon, underscoring the strong edaphic control on nitrogen cycling. Together, these findings reveal coherent geographic structures in both isotopic systems and highlight their complementary potential for ecological interpretation and for developing robust, science-based tools for timber provenance across the Amazon.

#### 4.1 Intra-site variability

 $\delta^{13}$ C and  $\delta^{15}$ N values in wood showed a large range at the Amazon-wide scale. The variation arises from the interplay between large-scale environmental controls, such as climate, water availability, soil composition, and locally operating ecological and physiological processes at the individual and species level (Leavitt and Roden, 2022; Savard and Siegwolf, 2022). Our dataset, which includes a broad taxonomic representation (Fig. S1), reflects part of the extraordinary tree diversity of the Amazon (Ter Steege et al., 2023). This exceptional biodiversity, bringing together species with contrasting functional strategies, inherently amplifies isotopic variability at local scales. In this sense, the high intra-site variability we observed in both isotopes is not surprising and confirms the importance of intra- and interspecific controls on isotopic composition in Amazonian woods, as also emphasized by (Batista et al., 2025). For example, when comparing two sites with multiple genera collected (Feijó and Itapuã do Oeste; Fig. 4a), we found systematic intra-genus and inter-genus differences in carbon isotopes.


**Figure 8.**  $\delta^{15}$ N isoscape and associated uncertainty. (a) Predictive isoscape of  $\delta^{15}$ N in Amazonian wood samples across 44 sites (circles), showing spatial variation in predicted values (‰) based on the final Random Forest model. (b) Map of associated predictive uncertainty expressed as the standard deviation (SD) of predicted  $\delta^{15}$ N values. Darker red areas indicate regions with higher model uncertainty.

Intra-genus variability of  $\delta^{13}$ C was not negligible (Table S2). In Itapuã do Oeste, *Handroanthus* exhibited the lowest dispersion (0.46 ‰), while *Cedrela* and *Dipteryx* displayed 0.92 ‰ and 1.2 ‰, respectively. This suggests that ecophysiological processes strongly contribute to local variability (McCarroll and Loader, 2004; Cernusak et al., 2013; Cernusak and Ubierna, 2022), partly reflecting interspecific differences within genera, as individuals were identified only at the genus level. This variance is likely explained by contrasting water-use efficiency (WUE) strategies (Ponton et al., 2001; Wittemann et al., 2024), driven by differences in stomatal conductance, ontogenetic stage, canopy position, and microhabitat conditions such as light






and water availability, which together affect photosynthetic rates and carbon isotope discrimination (Farquhar et al., 1982; Camargo and Marenco, 2011; Salmon et al., 2011; Van Der Sleen et al., 2014; Brienen et al., 2017; Cernusak and Ubierna, 2022). In *Cedrela*, experimental evidence shows anatomical plasticity in response to climatic variation (Ortega-Rodriguez et al., 2024). Studies in the Peruvian Andes reveal that species of this genus adjust xylem traits, such as vessel diameter and density, under drier conditions, reducing cavitation risk and enhancing hydraulic safety at the expense of efficiency (Rodríguez-Ramírez et al., 2022). These adjustments vary among species: *C. fissilis* shows greater recovery capacity after droughts, whereas *C. nebulosa* is less resilient, helping to explain part of the intra-genus variability observed in our data.

From a functional perspective, the observed hierarchy (*Handroanthus* > *Dipteryx* > *Cedrela* ) corresponds to the differences in water-use efficiency (WUE) observed between these genera and is commonly associated with other functional traits, such as wood density and xylem anatomy (Chave et al., 2009; Fichtler and Worbes, 2012; Hu et al., 2024).

Beyond anatomical considerations, the higher  $\delta^{13}$ C values in *Handroanthus* compared to *Cedrela* also reflect their contrasting ecological positions. *Handroanthus* and *Dipteryx* are typically emergent in terra firme forests, reaching heights above 40 m (Ribeiro et al., 1999), whereas *Cedrela* usually remains within the upper canopy. Taller trees in Central Amazonia exhibit greater stomatal density and stricter stomatal control to sustain water transport along longer hydraulic pathways (Camargo and Marenco, 2011; Brienen et al., 2017). These structural and ecological differences likely explain the consistently higher  $\delta^{13}$ C values and greater WUE in *Handroanthus*, reinforcing that structural and physiological traits together determine part of the isotopic composition of Amazonian wood (Batista et al., 2025).

For nitrogen isotopes, individuals from Itapuã do Oeste ( $5.66 \pm 1.19 \%$ ) consistently exhibited higher  $\delta^{15}$ N values than those from Feijó ( $2.25 \pm 1.67 \%$ ; Fig. 4b), highlighting strong inter-site differences (Nardoto et al., 2014). The intra-genus variability was generally lower at Itapuã do Oeste, except in *Cedrela* (SD  $\pm 1.66 \%$ ; Table S2), consistent with previous observations of high isotopic heterogeneity in *terra firme* forests of the Amazon (Nardoto et al., 2008). While  $\delta^{15}$ N values reveal clear spatial patterns at the regional scale, local variability can be large, as individuals may exploit distinct nitrogen sources or respond to micro-edaphic and hydrological variation depending on age, canopy position, or successional group. Systematic inter-genus differences were also observed, particularly in Feijó, with *Handroanthus* showing more positive and *Cedrela* and *Dipteryx* more negative  $\delta^{15}$ N values. This difference may reflect the edaphic conditions of Feijó, where the predominance of organic over mineral N creates a scenario in which species exploit distinct nitrogen sources (Michelsen et al., 1996; Craine et al., 2015). *Cedrela* and *Dipteryx* suggest greater use of surface organic N, typically characterized by lower  $\delta^{15}$ N values, while *Handroanthus* relies more on  $\delta^{15}$ N values, while handroanthus relies more on  $\delta^{15}$ N values, supporting previous observations that Amazonian legumes generally do not engage in significant N2 fixation (Högberg, 1997; Ometto et al., 2006; Nardoto et al., 2008, 2014).

Overall,  $\delta^{13}$ C variability was mainly structured at the taxonomic level, reflecting functional differences among genera, whereas  $\delta^{15}$ N variability was primarily driven by site-specific factors, with consistently higher values in Itapuã do Oeste compared to Feijó. These patterns are consistent with the findings of (Martinelli et al., 2021), who analyzed the foliar isotopic



composition of 4,205 leaves from evergreen forests across major Brazilian biomes. Together, these results highlight the reduced discriminatory power of carbon and nitrogen isotopes at the individual tree or intra-site level, where high variability obscures provenance signals and limits their effectiveness for fine-scale provenancing.

At broader spatial scales, both (Batista et al., 2025) and our study demonstrate that inter-site variance is consistently greater than intra-site variability, reflecting the influence of large-scale environmental gradients. Building upon these patterns, our results further clarify the underlying drivers, showing that δ<sup>13</sup>C variability is primarily associated with climatic conditions, particularly mean relative humidity (≥ 70 % vs. < 70 %), whereas δ<sup>15</sup>N reflects contrasting geological domains between Precambrian shields and Phanerozoic sedimentary basins (Fig. 3). This geographic structuring of carbon and nitrogen isotope patterns across the Amazon supports the view that these isotopes could provide useful information for applications in illegal wood provenance.

# 4.2 Relative humidity as the primary driver of $\delta^{13}$ C in Amazonian wood

The  $\delta^{13}$ C dataset reveals a well-defined spatial gradient across the Amazon that follows patterns of relative humidity and WUE. Despite the ecological and climatic heterogeneity of the region, the Random Forest model showed robust performance  $(R^2 \approx 0.60; \text{RMSE} \approx 0.59 \%)$  and identified mean annual relative humidity and mean annual vapor pressure as the main predictors (Fig. 5a, b). The isotopic trends broadly follow the moisture transport across the Amazon Basin, with vapor from the Atlantic accumulating against the Andean barrier (Salati et al., 1979). In the northwest, annual precipitation often exceeds 3,000 mm and relative humidity approaches 90 %. Dominguez et al. (2022) showed that up to 30 % of precipitation derives from local vapor recycling, which is more intense in the west and southwest, while Shi et al. (2022) confirmed that these regions are strongly controlled by abundant rainfall, favoring wood with more negative  $\delta^{13}$ C values. In the south, southeast, and far north, precipitation is lower, and relative humidity can be as low as 40–60 % (Marengo et al., 2018; Espinoza et al., 2019; Marengo et al., 2024; Aprile et al., 2024). The water balance depends more on evapotranspiration, reinforcing the climatic vulnerability of these areas and resulting in less negative  $\delta^{13}$ C values in trees.

In shield regions, annual precipitation is lower (1,600–2,200 mm) and the dry season is prolonged, with rainfall reduced to 20–40 mm month<sup>-1</sup> and relative humidity between 40–60% (Aprile et al., 2024). This water deficit leads to less negative  $\delta^{13}$ C values. The case of Roraima is illustrative: the north and east, dominated by savannas and open forests (Barbosa et al., 2007), show low annual precipitation and a well-defined dry season (monthly minima close to 30 mm) with higher  $\delta^{13}$ C values, whereas the west and southwest of the state are connected to the humid regime of the northwest, receiving stronger influence from vapor recycling and displaying more negative  $\delta^{13}$ C values (Barni et al., 2020). Within the same Guiana Shield, Amapá showed relatively less negative  $\delta^{13}$ C values. Despite high mean annual precipitation (Aprile et al., 2024), Marengo et al. (2001) highlighted that the eastern Amazon has a more marked dry season, with a later onset and earlier end of rainfall compared to the central and northwestern basin. This stronger seasonality may result in lower atmospheric saturation, reflected in lower mean relative humidity and higher vapor pressure deficit, conditions consistent with the isotopic results observed in Amapá.

The link between  $\delta^{13}$ C values in trees and atmospheric vapor circulation reflects underlying ecophysiological processes. Vapor pressure deficit (VPD), defined as the difference between the saturation vapor pressure ( $e_{max}$ ) and the actual vapor



pressure of the air  $(e_a)$ , is strongly correlated with relative humidity  $(r_{\rm H}=e_a/e_{\rm max};\,r\approx-0.96,{\rm Pearson})$  and is recognized as the main driver of stomatal conductance and  $\delta^{13}{\rm C}$  variation in  ${\rm C}_3$  plants (Cernusak et al., 2013; Novick et al., 2016). In our model, both mean annual relative humidity and mean annual vapor pressure were selected as predictors. Although they are physically related, relative humidity expresses the degree of air saturation, while vapor pressure captures absolute atmospheric moisture. The combination of these two variables allows the model to better represent spatial gradients in air humidity and its influence on  $\delta^{13}{\rm C}$  discrimination through VPD-related processes.

In practice, these two predictors are highly related to VPD. In regions such as the Solimões–Amazon corridor, extensive floodplains and closed canopy maintain air close to saturation, reducing VPD, increasing relative humidity, favoring greater stomatal opening, and enhancing discrimination against  $^{13}$ C, which results in more negative  $\delta^{13}$ C values (Farquhar et al., 1989; Lloyd and Farquhar, 1994; Cernusak et al., 2013). In the south and southeast, higher temperatures increase the air's saturation capacity, and relative humidity drops sharply during the dry season, intensifying evaporative demand, increasing stomatal limitation, raising VPD, and leading to less negative  $\delta^{13}$ C values. The partial dependence plots confirm this complexity:  $\delta^{13}$ C increases to a maximum under intermediate conditions of mean annual relative humidity ( $\sim$ 63–68%) and mean annual vapor pressure ( $\sim$ 2.7–2.85 kPa), when evaporative demand is intense, but decreases again under very humid conditions ( $r.hurs\_mean > 70$ % and r.vapor > 2.85 kPa), when the atmosphere approaches saturation and discrimination increases once more (Fig. 5b, c). The mean annual relative humidity  $\times$  mean annual vapor pressure interaction surface reinforces this pattern, with maxima under intermediate conditions and minima under saturated atmospheres, particularly along the humid Solimões–Amazon corridor (Fig. 5c). More negative values were also observed under low humidity and vapor conditions, which lack physiological support from the  $c_i/c_a$  relationship (Farquhar et al., 1989) and may be associated with the lower density of observations in these ranges of the gradient (Fig. 5b).

The highest modeling uncertainty for carbon isotopes is concentrated in the east, north, and in transition zones between the humid Solimões–Amazon corridor and less saturated areas (Fig. 6b), reflecting strong gradients of humidity and vapor (Fig. S4). Along these boundaries, small atmospheric variations generate more variable ecophysiological responses, increasing the dispersion of predictions. In contrast, the southern and southeastern Amazon display lower uncertainty, reflecting more homogeneous hydroclimatic conditions that are well represented by the covariates. Part of the uncertainty also stems from the low sampling density in the east and north, as well as intra-site variability linked to taxonomic differences. This highlights the need to expand the sampling network, particularly in transition regions, with more sites and replicates, or by focusing on taxonomically related species to reduce non-climatic variability.

## 4.3 Edaphic factors as the primary driver of $\delta^{15} N$ in Amazonian wood

The spatial distribution of  $\delta^{15}$ N is not predominantly influenced by climatic gradients but rather by edaphic factors. Variations between sedimentary and crystalline regions, associated with the quantity and quality of organic matter and the vigor of primary production, are encapsulated by three principal predictors: mean geological age, soil organic carbon, and net primary productivity. These variables emerged as the strongest descriptors of the observed  $\delta^{15}$ N pattern, with the Random Forest model showing robust performance ( $R^2 = 0.67$ ; RMSE = 1.10 ‰) (Fig. 7). Consistent findings were reported by Sena-Souza et al.





(2020) in their modeling of the continental distribution of soil  $\delta^{15}$ N across South America, where soil organic carbon and net primary productivity were identified as some of the most significant predictors, alongside edaphic and climatic variables, highlighting the critical role of these factors in shaping isotopic patterns associated with nitrogen cycling.

In our analytical model, the most significant predictor of  $\delta^{15}$ N variation was identified as the mean geological age (Fig. S6a), which closely mirrored the spatial distribution observed in the  $\delta^{15}$ N isoscape (Fig. 8a). Within the Amazon, the geological land-scape is structured into two principal domains: ancient crystalline formations, such as the Brazilian and Guiana shields, and younger Cenozoic sedimentary regions (Martinelli et al., 2025). The partial dependence plot (Fig. 7b) shows a gradual increase in  $\delta^{15}$ N with geological age, stabilizing at higher values in substrates older than  $\sim$ 1500 Ma. Consistent with this interpretation, Quesada et al. (2011) demonstrated that old cratonic surfaces are dominated by deeply weathered soils, particularly Ferralsols and Acrisols, whereas younger sedimentary surfaces sustain relatively young and less developed soils, such as Cambisols and Fluvisols. At the local scale, Nardoto et al. (2008) also reported higher foliar  $\delta^{15}$ N in clay-rich, highly weathered Oxisols (Ferralsols), where iron- and aluminum-oxide-rich soils favor denitrification and fractionating nitrogen losses. Together, these lines of evidence support the view that ancient, nutrient-depleted soils consistently sustain higher  $\delta^{15}$ N than younger, less developed sedimentary lowlands. Remarkably, similar trajectories have only been described in short-range soil chronosequences from the Hawaiian Islands, where foliar and soil  $\delta^{15}$ N values increase with substrate age as nitrogen availability peaks and fractionating losses accumulate, before stabilizing in very old, phosphorus-limited systems (Crews et al., 1995; Vitousek et al., 1995; Vitousek and Farrington, 1997). Our findings extend this pattern to a regional scale, demonstrating for the first time a consistent association between geological age and  $\delta^{15}$ N across the Brazilian Amazon.

This Amazonian dichotomy likely reflects regional differences in bedrock porosity, bedrock mineralogy, and soil type propagating in differences in water-holding capacity, drainage, and soil redox conditions (Huscroft et al., 2018). Soils in sedimentary lowlands likely experience more pronounced moisture fluctuations due to shallow water tables and seasonal saturation, whereas shield areas remain more stable. Experimental evidence from tropical soils demonstrates that such redox fluctuations can strongly structure microbial communities and nitrogen cycling processes (Pett-Ridge and Firestone, 2005). These patterns can be interpreted as background controls on soil hydrology, ultimately influencing nitrogen cycling pathways.

In the Amazon, younger sedimentary units, often derived from Andean erosion or associated with contemporary alluvial deposits, are typically located in lowlands with porous bedrock, shallow water tables, and seasonal saturation—conditions that favor hydrological fluctuations and create active redox environments. In such floodplain and alluvial settings, soils are often waterlogged and subject to denitrification and leaching losses. Nevertheless, vegetation  $\delta^{15}$ N values tend to remain relatively low because strong hydrological connectivity and continuous inputs of allochthonous nitrogen from upstream floods and sediments replenish the soil pool with isotopically lighter nitrogen sources, including substantial contributions from biological N<sub>2</sub> fixation (Martinelli et al., 1999; Hedin et al., 2009; Nardoto et al., 2008).

By contrast, the ancient crystalline basement of the Amazonian craton exhibits lower bedrock porosity, enhanced drainage, and reduced susceptibility to saturation (Gleeson et al., 2014; Huscroft et al., 2018). In these upland, well-drained settings, external nitrogen replenishment is minimal, and the nitrogen cycle depends strongly on the long-term internal recycling of organic matter (Martinelli et al., 1999; Davidson et al., 2007). Over long pedogenic timescales, the absence of significant






new inputs, combined with continuous microbial transformations, promotes fractionating nitrogen losses—such as gaseous emissions from volatilization of NH<sub>3</sub> and denitrification—that preferentially remove  $^{14}$ N and progressively enrich the residual nitrogen pool in  $^{15}$ N (Martinelli et al., 1999; Davidson et al., 2007; Nardoto et al., 2008, 2014). Consequently, upland cratonic regions display higher  $\delta^{15}$ N values than fluvial lowlands, reflecting long-term edaphic control, a pattern clearly identified in our  $\delta^{15}$ N isoscape.

In Amazonia, soils are highly heterogeneous (Richter and Babbar, 1991; Quesada et al., 2011) and shape the distribution and dynamics of tropical rainforest biodiversity (Cámara-Leret et al., 2017; Figueiredo et al., 2018; Schaefer et al., 2008; Tuomisto et al., 2003). Fittkau et al. (1975) distinguished three major geochemical domains: western Amazonia, with young and nutrient-rich soils of Andean origin; central Amazonia, with nutrient-poor and highly weathered soils; and the peripheral northern and southern regions, with intermediate soils derived from cratonic rocks. Recent maps of SB (sum of exchangeable bases), such as that presented by Zuquim et al. (2023), largely confirm this view but also reveal marked local heterogeneity, resulting from processes such as depositional and erosional rearrangements driven by fluvial dynamics, sedimentation under (semi)marine conditions, and *in situ* weathering (Rossetti et al., 2005; Hoorn et al., 2010; Quesada et al., 2011; Higgins et al., 2011).

Our  $\delta^{15}N$  isoscape broadly follows these geochemical patterns but also highlights relevant deviations. Central Amazonia exhibited relatively low  $\delta^{15}N$  values, in contrast to the expectation of higher values typically associated with the ancient, nutrient-depleted soils that dominate this region (Ometto et al., 2006). These lower values are closer to those recorded in areas under stronger Andean influence, such as Acre and southern Amazonas, rather than the higher values characteristic of cratonic regions. This behavior confirms the west–east increase in  $\delta^{15}N$  described in previous studies and underscores the role of regional hydrology in modulating this gradient (Nardoto et al., 2008, 2014; Sena-Souza et al., 2020).

Although our model predicts higher  $\delta^{15}N$  values in the east, a band along the axis of the Amazon River displays relatively lower values, reflecting the strong hydrological connectivity associated with the river and its major tributaries. Nonetheless, some sites retain high  $\delta^{15}N$  values, such as Belterra in the Santarém region, already recognized in the literature as an area of high plant  $\delta^{15}N$  (Ometto et al., 2006; Nardoto et al., 2014). In this case, the relatively elevated terrain and well-drained soils differentiate the area from adjacent floodplains, favoring more intense mineralization and fractionating nitrogen losses, which enrich the residual nitrogen pool in  $^{15}N$ . Moreover, Santarém is comparatively drier than the Central Amazon, a condition that further enhances organic matter mineralization, and its dominance of clay-rich soils likely contributes to consistently higher  $\delta^{15}N$  values.

The second most important predictor in the model was Soil Organic Carbon (SOC), which acts as an indirect regulator of  $\delta^{15} N$  by reflecting both the quantity of accumulated organic matter and its stage of decomposition (Quesada et al., 2010). In Amazonian soils, high SOC levels are typically found in geologically younger and less weathered regions, often under strong hydrological influence. In these environments, flood–ebb dynamics enhance the input of fresh organic matter and delay nitrogen mineralization. Under such conditions, leaching and denitrification become more intense, leading to lower  $\delta^{15} N$  values (Amundson et al., 2003; Houlton et al., 2006; Pérez et al., 2006). Conversely, low SOC levels are more common in geologically older, well-drained areas, where mineralization is accelerated and the residual nitrogen pool tends to exhibit higher  $\delta^{15} N$  values





(Nardoto et al., 2008; Craine et al., 2015). This relationship is clearly illustrated in the interaction plot (Fig. 7c), which shows higher  $\delta^{15}$ N values under the combined conditions of older bedrock and low SOC, whereas younger, carbon-rich soils tend to sustain lower values.

The least important variable was Net Primary Productivity (NPP). Although its overall contribution to the model was small, it significantly reduced prediction error. Previous studies in the Amazon indicate that NPP increases in more fertile soils, particularly those enriched in phosphorus and with higher foliar nitrogen (Aragão et al., 2009; Quesada et al., 2011). In such conditions, greater nutrient availability promotes vegetation growth and aboveground biomass accumulation, indirectly influencing nitrogen cycling. This effect was apparent only at a few sites, which explains the minor importance of NPP in the overall model.

The model shows a high degree of uncertainty in the Guiana Shield, reflecting the scarcity of samples and poor training in those regions. Elevated uncertainty persisted in Rondônia in the south despite good coverage, likely because the combination of contrasting soils and geomorphological heterogeneity generates hydrological microenvironments that increase isotopic variability. This pattern is consistent with the isoscape (Fig. 8a), which shows a mosaic of high and low  $\delta^{15}$ N values. In contrast, the south and southeast exhibit low uncertainty, consistent with more homogeneous environmental conditions. These results highlight the importance of expanding the sampling network, with priority given to critical regions such as the cratons, rock–sediment transition zones, and the Andean foreland. As with  $\delta^{13}$ C, this expansion should include a greater number of sites and replicates per locality, as well as the selection of taxonomically related genera or species to reduce non-climatic variability.

## 4.4 Limitations and perspectives

Although the models demonstrated good predictive performance, several important limitations remain, largely associated with the uneven sampling network across the basin. Uncertainty is particularly high in the cratons, in rock—sediment transition zones, and along the Andean foreland, where isotopic gradients are poorly represented by the available data. Inter-specific differences may also inflate the apparent spatial variability, reinforcing the value of constructing or calibrating isoscapes at least at the genus level for specific applications.

To increase the predictive power of these isoscapes, it is essential to expand the representativeness of the Amazon through new sampling efforts, especially in regions that remain underrepresented. It is also important to consider the taxonomic proximity among species in the sampling design, which may help reduce model uncertainties. Building a collaborative network, together with standardized sampling protocols, is crucial to ensure that all studies target the same radial position, allowing for more consistent comparisons and stronger discussions.

Even with these limitations, it is important to apply the models in real-world contexts and to assess their usefulness, costs, and potential integration into existing monitoring and enforcement systems. Overall, the combination of C and N isoscapes with geospatial predictions represents a promising and cost-effective pathway to increase the robustness of provenance assignments.


#### 5 Conclusions

For the first time at the scale of the entire Amazon basin, we mapped  $\delta^{13}C$  and  $\delta^{15}N$  variations in wood. Our results show that these isotopic systems are informative for ecological processes and also hold practical value for timber traceability. Carbon isotopes primarily reflect broad climatic gradients, whereas nitrogen isotopes capture more localized edaphic processes. Despite substantial intra-site variance, the combination of these isotopes provides a preliminary screening method that can identify potentially suspicious timber batches.

Although on their own they do not allow provenance at the level of individual logging sites, these isotopes establish a solid scientific foundation for regional-scale monitoring. This approach is particularly relevant in the Amazon given its vast territorial extent and the logistical complexity of enforcement. This study also advances the perspective of multi-isotope integration in which low-cost carbon, nitrogen, and oxygen analyses can serve as first-line screening followed by more precise and expensive systems such as sulfur, strontium, or genomic tools for definitive verification.

By demonstrating the feasibility of basin-wide wood isoscapes and applying scalable machine-learning approaches, this work contributes to tropical forest monitoring and to understanding forest functioning in relation to biogeochemical processes. These predictive frameworks can be applied in other tropical regions and offer a pathway toward global applications. The practical demands of forest governance make the isoscapes presented here a valuable complement to certification systems and sustainable management, directly strengthening efforts to secure a more reliable legal timber trade and to combat illegal logging in the Amazon.

Code and data availability. All code and data supporting this study are available in the Open Science Framework (OSF) repository: https://osf.io/u5rws/overview. The repository is publicly accessible.

Author contributions. IMSS conceived and designed the study, curated and analyzed the data, conducted the investigations and methodological procedures, and wrote the original draft of the manuscript as well as its subsequent revisions. LAM contributed to the conceptualization, investigation and methodological development, founded the project, provided resources, supervised the research, and contributed to the writing and revision of the manuscript. BH contributed to the formal analysis and to the writing and revision of the manuscript. ACGB, MGSA, ALG, SP, GML, DORR, DJA, FJVC, GBN, ATB, GAP, JPSS and VEC contributed to the writing and revision of the manuscript. PG, MTF,
 NH and ACB provided wood samples and contributed to the writing and revision of the manuscript. CPB contributed to the conceptualization, data curation, formal analysis, investigation, methodology and supervision, and wrote and refined all versions of the manuscript.

Competing interests. The authors declare that no competing interests are present.

Disclaimer. None.

Acknowledgements. This work was made possible through the collaboration of Forest Management – INPA, the Dendrochronology Laboratory and Identification Laboratory (LAIM) – ESALQ/USP, the Dendrochronology Laboratory – UNICAMP, and the companies Mil Madeiras Preciosas Ltda. and Teak Resources Company, which generously provided wood samples. We thank Aparecido Candido Siqueira, technician at the LAIM, for assistance in sample preparation, and the team at the Isotopic Ecology Laboratory (LEI) – CENA/USP, including Fabiana C. Fracassi Adorno, Gustavo Gobert Baldi, and Isadora S. Ottani, for support in analytical processing, as well as Sarah Lima for administrative assistance. We are also grateful to all other colleagues at CENA/USP who contributed to this project.

Financial support. This work was supported by multiple funding agencies and institutions. Research activities were funded by CAPES (Academic Cooperation Program in Public Security and Forensic Sciences – PROCAD), INCT–CNPq (Forensic Metrology and Traceability in Agro-Environmental Quality – MRFor), and The Nature Conservancy Brazil (TNC) in partnership with Google. This study was financed in part by the São Paulo Research Foundation (FAPESP; grant nos. 2023/13568-7 to I. M. Souza-Silva and 2018/01847-0 to P. Groenendijk) and by the Brazilian National Council for Scientific and Technological Development (CNPq; grant nos. 140304/2022-3 and 306333/2024-4). A. T. Brunello was supported by a CNPq postdoctoral fellowship (process no. 157802/2025-6), V. E. Costa grateful for the research productivity scholarship (CNPq, 316659/2021-5), and C. P. Bataille was funded by the Purdue University College of Agriculture startup fund.

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
