# Peer review of "Machine-learning models of $\delta^{13} C$ and $\delta^{15} N$ isoscapes in Amazonian wood"

_EGUsphere, 2025_

## Author Comment (AC1)

**Reply to comments on "Machine-learning models of δ¹³C and δ¹⁵N isoscapes in Amazonian wood" by Souza-Silva et al.**

We are grateful to the editor and reviewers for their thorough assessment of our manuscript. Their detailed feedback and thoughtful suggestions have been instrumental in refining the study. We have carefully addressed all comments and outline our responses below.

**Referee 1**

General comments:

The manuscript describes an extensive sampling of ¹⁵N and ¹³C in wood across the Amazon. A major goal of the project is to investigate whether isotopes can be used to provenance lumber, with relevance to monitoring illegal logging. Towards this goal, random forests models are fitted to the isotope datasets using a large suite of assembled spatial, ecological, pedological, and climatological covariates. RF models are subsequently used to produce isoscapes: spatially resolved isotope predictions across the Amazon. Patterns of isoscape variation are discussed in the context of various ecological factors.

Major comments :

1. Given the collection of two independent isotope signals, I wonder at the missed opportunity here to generate a combined isoscape of ¹³C and ¹⁵N variation. It seems the combined variation in both isotopes would be more spatially variable given distinct controls. Mapping the product of the two, or some other decomposition or principal component, could actually generate a more useful metric.

   **Response:** We thank the reviewer for this insightful suggestion. We concur that analyzing δ¹³C and δ¹⁵N together could enhance spatial contrast for geographic attribution, especially since these isotopes are influenced by largely distinct environmental drivers. Composite metrics and multivariate approaches, including principal component analysis, can indeed play a valuable role in provenance-oriented analyses. Nevertheless, the primary objective of this study is to establish isotope-specific reference isoscapes for Amazonian wood and to investigate the environmental and biogeochemical controls on each isotope system, rather than to perform formal sample assignment or to construct a unified provenance index.

Although both $\delta^{13}C$ and $\delta^{15}N$ exhibit coherent basin-scale spatial patterns, their pronounced within-site variability currently constrains their standalone applicability for fine-scale provenance assignment. We address this limitation in the Conclusion, where we highlight that $\delta^{13}C$ and $\delta^{15}N$ should primarily be considered as a solid basis for regional-scale monitoring and for initial screening of potentially suspicious timber, rather than as conclusive indicators of origin. In this sense, these isotopes are best interpreted as complementary evidence that can inform provenance assessments, recognizing that more accurate assignments will likely depend on incorporating additional isotopic tracers that reflect independent environmental signals. We therefore recognize the reviewer's recommendation as a promising direction for future work. Subsequent studies could embed these isotope-specific isoscapes within multivariate provenance frameworks, potentially alongside other isotopic systems, such as sulfur or strontium, to enhance assignment accuracy and refine geographic discrimination.

2. The study effort, particularly around sample collection, is ambitious and admirable. I agree that the current results are not particularly useful for provenance verification. But perhaps combined with other isotope maps (strontium?) could be in the future.

   **Response:** We thank the reviewer for this positive assessment of the sampling effort. We agree that, on their own, the $\delta^{13}C$ and $\delta^{15}N$ isoscapes presented here are not intended for definitive provenance verification at fine spatial scales. As emphasized in the manuscript, these isotope systems are best suited for regional-scale monitoring and preliminary screening. We also agree that integrating carbon and nitrogen isoscapes with other isotopic systems represents a promising avenue for further research. In this context, a strontium isoscape for the Amazon basin has recently been developed (Martinelli et al., 2025) and provides an independent geochemical framework that could, in future studies, be combined with the $\delta^{13}C$ and $\delta^{15}N$ isoscapes presented here within a multi-isotope provenance approach.

3. The machine-learning methods appear robust, but I would be curious to see added some additional discussion around systematic bias of the RF models, which appear to underpredict high values and overpredict low values systematically.

**Response:** We thank the reviewer for this observation. We agree that the Random Forest models show a tendency to underpredict higher values and overpredict lower values, a pattern that is evident in the observed versus predicted relationships (Figs. 5a and 7a) and is characteristic of ensemble tree-based regression methods. To address this point, we have added an explicit discussion of this behavior in the Discussion section, as the beginning paragraph of Section 4.4 (Limitations and perspectives). The following paragraph has been inserted: *"Although a systematic model bias is observed, characterized by a tendency to overpredict low isotope values and underpredict high values, the overall performance of the Random Forest models remains robust at the basin scale. This conservative smoothing behavior reflects a regression-to-the-mean effect inherent to ensemble tree-based methods and is amplified when data density is highest in the intermediate range of values and sparser at the extremes. Importantly, this behavior does not indicate a spatially structured bias and instead supports stable basin-scale predictions by reducing overfitting"*.

Minor comments:

- Line 123. Perhaps worth explaining how a sample was collected 215 km from a road, and how frequent such samples are in the dataset. I am imagining this was transported by boat? Then could redefine as distance from road or river?

  **Response:** We thank the reviewer for this request for clarification. The reported maximum distance of 215 km does not represent the distance of a sample from a road or river, but rather the maximum pairwise distance between individual trees within a single sampling site, based on their geographic coordinates. This value corresponds to a single individual that was located approximately 215 km from the other sampled trees at the same site, resulting in an unusually large intra-site distance. This situation reflects the vast territorial extent of some Amazonian municipalities, where a single "site" may encompass a very large area. Importantly, this is an isolated case in the dataset and involves only one individual. For the vast majority of sites, sampled trees are clustered within a few kilometers of each other, with a median intra-site distance of less than 1 km, which better represents the typical spatial configuration of our sampling design.

- Fig 1. This figure could be made more useful by somehow depicting sample size at each site, either as point size / color or with a number. This would allow visualization of the distribution of sampling intensity across the region.

**Response:** We thank the reviewer for this helpful suggestion. We agree that visualizing sampling intensity across sites provides important context for interpreting the spatial distribution of the dataset. Figure 1 has therefore been revised to explicitly display sample size at each site. Sampling intensity is now represented by varying point sizes, allowing readers to readily assess the number of sampled individuals per site across the Amazon region. More detailed and precise information on the number of individuals sampled at each site is also provided in Table 1. This modification enhances the interpretability of the figure while preserving its original purpose of illustrating the geographic extent of the sampling effort.

[Figure]

**Figure 1.** Sampling sites across the Brazilian Amazon. (a) South America, with emphasis on the Brazilian Amazon region. (b) Distribution of sampled individuals across 47 sites in the Brazilian Amazon. Numbers correspond to site identifiers listed in Table 1. The map highlights the states where sampling took place, with grey lines indicating state boundaries. Yellow symbols represent sampled sites, while blue symbols (sites 20, *Feijó*, and 24, *Itapuã do Oeste*) indicate the locations analyzed in detail in this study. Symbol size is scaled according to the number of individuals sampled per site (see legend). South America boundaries were obtained from Natural Earth (2023), and the Amazon biome polygon from Assis et al. (2019).

- L349. You might refer to this as bias. $^{13}C$ isoscape also exhibits the same bias.

   **Response:** We thank the reviewer for this clarification and have revised the text around L349 to explicitly refer to this behavior as a systematic bias, noting that it is common to both the δ¹³C and δ¹⁵N isoscapes. We decided to insert a new paragraph after L288 (line according to the revised version of the manuscript) to demonstrate that this behavior represents a bias. "*The scatter plot of observed versus predicted δ¹³C values reveals an approximately linear relationship (Fig. 5a). However, the fitted regression line (black) deviates from the identity line (dashed), indicating a tendency to overestimate δ¹³C at the lower end of the distribution (more negative δ¹³C values) and to underestimate at the upper end. This systematic deviation can be referred to as a model bias, characterized by regression toward the mean. The highest density of points occurs within the intermediate δ¹³C range (−28 to −27 ‰), with fewer observations toward both extremes.*" In addition to this paragraph, we revised the text at L319 (line according to the revised version of the manuscript), where the original paragraph has been replaced by the following: "*The scatter plot of observed versus predicted δ¹⁵N values demonstrates a predominantly linear trend across the entire range of variation (Fig. 7a). A systematic deviation is apparent, consistent with the conservative smoothing observed for δ¹³C, with higher δ¹⁵N values being underestimated and lower values being overestimated. The majority of data points are densely populated in the intermediate sector, specifically between 3.5 ‰ and 5.5 ‰, whereas fewer observations are located at the extremes of the distribution*".

- L440. Many sites have δ¹³C variation of around 3-4 per mil. So while I suppose this is true, inter-site variability is not much greater than within site.

   **Response:** We thank the reviewer for this careful observation and agree that many sites exhibit substantial within-site δ¹³C variability, often on the order of 3–4‰. Such pronounced intra-site variability is expected in Amazonian forests and reflects strong ecological, physiological, and taxonomic heterogeneity operating at local scales. As discussed in the manuscript, this local-scale variability coexists

with broader-scale environmental gradients that structure $\delta^{13}C$ variation across the basin. Importantly, a similar magnitude of intra- and inter-site variability does not imply the absence of meaningful spatial differentiation. Previous studies have shown that, despite high within-site heterogeneity, between-site differences in $\delta^{13}C$ remain detectable and informative at regional scales. For example, Batista et al. (2025) documented pronounced intra-site variability in Amazonian tree-ring $\delta^{13}C$ while still identifying a dominant inter-site component. Accordingly, our results support the interpretation that $\delta^{13}C$ isoscapes are most informative for regional-scale patterns rather than fine-scale site discrimination.

**References**

Batista, A. C. G., Silva, I. M. S., Silva Araújo, M. G. D., Amorim, D. J., Nardoto, G. B., Costa, F. J. V., Higuchi, N., Tomazello-Filho, M., Barbosa, A. C., Costa, V. E., Ponton, S., and Martinelli, L. A. (2025). *Within- and between-site variability of $\delta^{18}O$, $\delta^{13}C$, and $\delta^{15}N$ in Amazonian tree rings: Climatic drivers and implications for geographic traceability*. **Forest Ecology and Management**, 597, 123168. https://doi.org/10.1016/j.foreco.2025.123168.

Martinelli, L., Bataille, C., Batista, A., Souza-Silva, I., Araújo, M., Abdalla Filho, A., Brunello, A., Tommasiello Filho, M., Higuchi, N., Barbosa, A., Costa, F., and Nardoto, G. (2025). *Bioavailable strontium isoscape for the Amazon region using tree wood*. **Forest Ecology and Management**, 594, 122963. https://doi.org/10.1016/j.foreco.2025.122963.

---

## Author Comment (AC2)

**Reply to comments on "Machine-learning models of δ¹³C and δ¹⁵N isoscapes in Amazonian wood" by Souza-Silva et al.**

We are grateful to the editor and reviewers for their thorough assessment of our manuscript. Their detailed feedback and thoughtful suggestions have been instrumental in refining the study. We have carefully addressed all comments and outline our responses below.

**Referee 2**

General comments

As a stable-isotope researcher with >15 years of experience, I read the work by Souza-Silva et al. on "Machine-learning models of δ¹³C and δ¹⁵N isoscapes in Amazonian wood" with great interest. The authors deliver an exceptionally valuable resource to the community: isotopic measurements for >550 trees distributed across 47 Amazonian sites, coupled with a robust Random-Forest framework that links wood δ¹³C and δ¹⁵N to 74 environmental predictors. The rationale is sound and the methodology state-of-the-art. I therefore strongly recommend its publication. Nevertheless, two issues merit attention before the manuscript reaches its full potential.

**Response:** We sincerely thank the reviewer for this very positive and thoughtful assessment of our work. We appreciate the recognition of the dataset, the methodological framework, and the potential value of these isoscapes for the community. Below we address the specific points raised.

Specific comments:

1. Basin-scale timber provenancing: although the δ¹³C and δ¹⁵N isoscapes represent a commendable advance, their ability to discriminate Amazonian from non-Amazonian timber (or other regions within the Amazonian) remains equivocal. The observed intra- and inter-site variability is sufficiently large that isotopic signatures from other tropical forests may fall within the Amazonian envelope. I therefore urge the authors to (i) incorporate a comparative dataset of timber collected outside the basin and (ii) provide quantitative classification metrics (sensitivity, specificity) to rigorously demonstrate—and transparently bound—the utility of the isoscapes for large-scale provenance assignment.

**Response:** We thank the reviewer for this important and carefully articulated comment. We acknowledge that comparative analyses involving timber collected both within and outside the Amazon basin constitute an ambitious and valuable approach for addressing questions related to inter-basin discrimination and global provenance frameworks. Nevertheless, the aim of the present study is not to test the discrimination between Amazonian and non-Amazonian timber, but rather to establish isotope-specific reference isoscapes for $\delta^{13}C$ and $\delta^{15}N$ in Amazonian wood and to investigate the environmental processes controlling their spatial variability across the basin. These isoscapes are intended to provide a reference framework that can support future timber traceability studies, while also advancing the understanding of biogeochemical dynamics across the Amazon region. Addressing point (i), namely the incorporation of non-Amazonian datasets, would therefore represent a distinct and substantially more demanding research objective, requiring a broader sampling effort and a fundamentally different experimental design, including harmonized external datasets and supervised classification protocols, which lie beyond the scope of this manuscript. With respect to point (ii), we recognize that $\delta^{13}C$ and $\delta^{15}N$ alone are insufficient for definitive provenance verification, particularly at fine spatial scales. As discussed in the manuscript, these isotopes are best viewed as complementary tracers that support regional-scale monitoring and preliminary screening. More robust provenance assignment is expected to arise from multi-isotope frameworks integrating tracers that convey independent environmental information, such as strontium and sulfur. In this context, a bioavailable strontium isoscape for the Amazon basin has recently been developed (Martinelli et al., 2025), providing a solid foundation for future multi-isotope provenance applications. Such frameworks will be better suited for the implementation of formal classification analyses and for the calculation of quantitative performance metrics (e.g., sensitivity and specificity), as suggested by the reviewer.

2. Conciseness and focus: the manuscript, in its present form, is unduly verbose. Lengthy expositions in the Introduction, Results, and Discussion dilute the key messages and obscure the novel contributions. I strongly recommend a comprehensive tightening of the text—prioritizing only the most essential

findings and their implications—to enhance clarity, readability, and impact for the broad readership of this journal.

**Response:** We thank the reviewer for this constructive recommendation. We fully agree that improved conciseness will strengthen the manuscript. In response, we have carefully revised the Introduction, Results, and Discussion to reduce redundancy, remove less essential narrative elements, and sharpen the focus on the most relevant findings and their broader implications.

**Reference**

Martinelli, L., Bataille, C., Batista, A., Souza-Silva, I., Araújo, M., Abdalla Filho, A., Brunello, A., Tommasiello Filho, M., Higuchi, N., Barbosa, A., Costa, F., and Nardoto, G. (2025). *Bioavailable strontium isoscape for the Amazon region using tree wood*. **Forest Ecology and Management**, 594, 122963. https://doi.org/10.1016/j.foreco.2025.122963.